# Tracing the origin of lithium in Li-ion batteries using lithium isotopes

Anne-Marie Desaulty [1 ✉], Daniel Monfort Climent [1], Gaétan Lefebvre[1], Antonella Cristiano-Tassi[2], David Peralta [3], Sébastien Perret[1], Anthony Urban[2] & Catherine Guerrot [1]

Rechargeable lithium-ion batteries (LIB) play a key role in the energy transition towards clean energy, powering electric vehicles, storing energy on renewable grids, and helping to cut emissions from transportation and energy sectors. Lithium (Li) demand is estimated to increase considerably in the near future, due to the growing need for clean-energy technologies. The corollary is that consumer expectations will also grow in terms of guarantees on the origin of Li and the efforts made to reduce the environmental and social impact potentially associated with its extraction. Today, the LIB-industry supply chain is very complex, making it difficult for end users to ensure that Li comes from environmentally and responsible sources. Using an innovative geochemical approach based on the analysis of Li isotopes of raw and processed materials, we show that Li isotope 'fingerprints' are a useful tool for determining the origin of lithium in LIB. This sets the stage for a new method ensuring the certification of Li in LIB.

[1] BRGM, F-45060 Orléans, France. [2] EDF, EDF R&D, 77818 Moret sur Loing, France. [3] Université Grenoble-Alpes, CEALITEN, 38054 Grenoble, Cedex 9, France. ✉email: am.desaulty@brgm.fr

ithium, hyped as the "white oil" (*petróleo blanco*) or the "white gold" of the 21st century, owes its outstanding economic success to its key role in the energy transition[1]. Historically, lithium has found wide use in ceramic, glass, steel, and chemical industries, as well as in medicine for treating bipolar disorders. Recently, however, the lithium market has become dominated by Li salts used in rechargeable batteries, which now consume ~65% of all lithium[2].

Lithium-ion battery (LIB) is the term used for a battery composed of multiple electrochemical cells, each of which has a lithium-metal-oxide-based positive electrode (cathode) and a negative electrode (anode, typically graphitic carbon active material), electronically separated by a thin porous plastic film (i.e., separator) which contains the non-aqueous electrolyte solution (general comprising LiPF6 as salt and organic carbonates as solvents), and electronic current collectors (general Cu at the anode and Al at the cathode) that connect the electrochemical cell to an external circuit containing the load to be powered.

LIBs are widely used in portable electronic devices (tablets and mobile phones), and increasingly in cordless electric tools, transportation applications (hybrid and electric vehicles, electric scooters, e-bikes), and stationary power storage for intermittent energy sources (solar or wind). Electrification of transport is becoming a top priority as part of the transition to a low-carbon future, in particular, to meet the targets of the Paris climate agreement of reducing carbon emissions by more than a third by 2030[1]. Several recent government initiatives incentivise or even compel car owners to switch to electric: Norway will ban the sale of petrol-powered cars by 2025, while the United Kingdom, Ireland, Germany and the Netherlands plan to do the same by 2030, and France by 2040[3]. The recent EU plan for tackling global heating proposes banning new internal-combustion engines by 2035[4]. The demand for lithium thus will continue to rise as long as LIBs are the primary power source for electric vehicles (EV). The annual quantity of lithium required should increase by a factor of 44 by 2030 (considering a hypothesis of 0.8 million tons of lithium carbonate in 2030) compared to 2017 production volumes, to satisfy further needs in the mobility sector[5].

Commercial LIB currently uses various cathode compositions including ~5–10% of lithium[6] obtained from lithium salts (lithium carbonate or lithium hydroxide), and different ratios of other metals. The electrolyte, composed of lithium hexafluorophosphate (LiPF$_6$) diluted in solvent (LiPF$_6$/1 mol/L), contains a negligible quantity of lithium compared to the cathode material. High-nickel cathode compounds, in particular lithium-nickel-manganese-cobalt oxide (Li(NiMnCo)O$_2$ or NMC), are the most-used cathode materials today for EV applications and stationary storage[6]. First-generation LIB cathodes contained nickel-manganese-cobalt in the proportion of 1:1:1 (often identified in industry jargon as NMC111 or NMC333). In order to increase energy density, the Ni:Mn:Co ratio has gradually shifted from 1:1:1 to 5:3:2 to 6:2:2 to 8:1:1 to reduce the amount of Co required[7]. Increasing adoption of higher-nickel cathode compounds has led to greater use of lithium hydroxide, leading to higher-quality cathode materials with a better cycle life and energy density[8]. According to earlier studies[6,9], NMC cathodes will represent between 60 and 90% of annual battery demand by 2030, the other battery cathode types being NCA (nickel-cobalt-aluminium) and LFP (lithium-iron-phosphate). In coming years, the main LIB cathode evolution will concern the presence and quantities of cobalt, nickel, manganese, aluminium (NCA batteries) or phosphorus (LFP), but lithium will remain an indispensable component.

From a lithium viewpoint, the LIB manufacturing supply chain is complex and separated into many stages, including mining, extractive and refining metallurgy, cathode active material synthesis, battery-cell manufacturing, and battery-pack assembly, which commonly are completed in different locations and countries.

Most mineral reserves occur in Chile (44%), Australia (22%), Argentina (9%), China (7%), and several other countries accounting for the remaining 18%[10]. Lithium resources are primarily divided into three categories[11]: (1) Brine is the main source of lithium with close to 60% of the global identified reserves. Among the brines, salars in the "lithium triangle" of Bolivia, Argentina, Chile, and in Qinghai province and the Tibetan region of China, hold most of the lithium-brine reserves. (2) Hard-rock lithium resources, i.e., lithium-rich pegmatite is the second source by the amount of available lithium. Recent estimates account pegmatites for ~30% of identified lithium reserves. Among the minerals containing lithium in pegmatites, spodumene (LiAlSi$_2$O$_6$) is the primary economic mineral[12]. Lithium hard-rock reserves are distributed around the world, the largest spodumene deposits occurring in Australia with major deposits in Canada and China[10]. (3) Sediment-hosted deposits (sometimes erroneously generalised as "clay") are the third source, representing less than 3% of global lithium resources. They consist of hectorite (McDermitt, USA, and Sonora, Mexico) and jadarite (Jadar, Serbia). Producing lithium from this source has so far proven difficult and costly, and hitherto no company has been able to produce commercial quantities from such deposits. In 2020, almost half (47%) of global lithium production came from Australian hard-rock deposits. Other main suppliers were Chile (21%), China (17%), Argentina (7%) and a group of countries including Zimbabwe, USA, Brazil and Portugal (7%)[10].

After mining, the next step in the supply chain is extractive and refining metallurgy, the processing and purification that transforms raw materials into high-purity lithium hydroxide or -carbonate. The world's lithium-refining capacity is concentrated in China, which supplies over half (53%) of global lithium salts, including most lithium hard-rock production[13], whereas Chile (33%) and Argentina (11%) dominate refined lithium capacity from brine operations[8].

The production of cathode active materials, the manufacturing of battery cells, and the assembly of battery packs as the final product, are the other steps in the LIB supply chain. The lithium-metal oxide for the cathode active material is mostly produced by speciality chemicals companies in China, Japan and South Korea, which deliver 86% of active material[14]. China is also a major player in Li-ion cell manufacturing with 66% of global cell production, other suppliers being South Korea and the United States, with 13% each[13]. For EVs, manufacturers design battery packs for specific models, and tend to assemble them near the vehicle assembly plant because of the cost of transporting the large and heavy battery packs[15]. China is the largest battery producer for EVs, followed by the United States and Germany[5].

The LIB life cycle does not stop there, and supplementary stages can take place. After the use in EVs, LIB life can be extended by repurposing them for less demanding applications, such as energy storage[14]. Even if this is not cost-effective today compared to primary resources, the lithium contained in battery cells can be recycled and reused for manufacturing new cathodes.

Another complication in the supply chain of the LIB industry is the fact that every consumer company deals with several suppliers, each of which can deal with multiple sub-suppliers in various countries. For instance, to supply a substantial portion of its lithium needs, the American company Tesla has signed contracts directly with Ganfeng Lithium, a Chinese lithium mining and refining company (see Methods section), which has several subsidiaries involved in the lithium industry in Australia, China, Argentina, Mexico and Ireland[16]. The contract gives Tesla assured access to lithium, but in practice, the raw material passes

through many other companies and processing steps before it makes it into a car. Panasonic and CATL, which assemble battery cells for Tesla, source cathode active material from various chemicals companies (Sumitomo, BASF Toda, Beijing Easpring, Ecopro, Johnson Matthey)[17], which themselves buy lithium from various refining and mining companies.

Because of this complex supply chain, ensuring that raw materials come from socially and environmentally responsible sources with a low-carbon footprint, is a complicated puzzle for end users. Although the lithium supply chain today is less problematic in terms of social and environmental risks than other battery metals, such as cobalt, lithium can be associated with various environmental and social impacts. With the increasing demand for lithium, the environmental and social impacts of mining tend to increase.

In Argentina, indigenous communities report that lithium operations on their lands threaten their survival and the exercise of their rights[18]. In Zimbabwe, where lithium exploitation is currently low (1%)[10], the illicit financial flow has already been identified in the lithium mining sector[19]. A recent study[20] on the life-cycle water-scarcity footprint showed that water use associated with lithium-brine mining in Chile and China, mainly through evaporative loss, can create a high risk of natural freshwater scarcity for humans and nature.

According to a comparative Life Cycle Assessment[21] between EV batteries with hard-rock or brine-based lithium, the environmental impact of hard-rock lithium processing, dominated by traditional sulphuric acid processing and melting of the rock, is higher in terms of acidification and global warming potential. A recent study showed that the $CO_2$ equivalent emissions from hard-rock lithium hydroxide production in Australia and refining in China are up to three times higher than those from brine production in Chile and Argentina[22]. Another recent study, based on Life-Cycle Analyses of battery-grade lithium salts produced from Chilean brine and from Australian spodumene processed in China, showed that the production of Li salts from brine-based lithium had less life-cycle greenhouse-gas emissions and freshwater consumption than lithium salts from rock-based lithium resources[23].

Major car manufacturers like BMW group, Tesla, and Volvo recently announced that they will increase the transparency of their supply chains for EV batteries, and ensure responsible and sustainable sourcing of raw materials[24,25]. Some companies (BASF, Volkswagen, Fairphone) have started a partnership for sustainable lithium mining in Chile[26]. Carmakers also explore the usefulness of blockchains for improving the scrutiny of supply chains. A blockchain is the control of chain-of-custody systems, based on the shipping documentation that is included in online databases, to allow real-time raw materials tracking and electronic tagging. However, document-based traceability systems can be falsified, and must be independently controlled and audited to provide credibility[27].

We propose here an innovative geochemical approach based on analytical fingerprints of lithium isotopes of raw and processed materials, to ensure the traceability of lithium in LIBs. This method helps verify and audit the blockchain, thus ensuring its control. It was developed for the coltan supply chain[28] and, more recently, for native gold from French Guiana[29]. Lithium (Li) has two stable isotopes, $^6Li$ and $^7Li$, with relative abundances of 7.6% and 92.4%, respectively. The Li isotope compositions ($\delta^7Li$) are reported as a classical δ-notation (parts-per-thousand, ‰) with the $^7Li/^6Li$ ratio relative to standard lithium (L-SVEC)[30]: $\delta^7Li = [(^7Li/^6Li)_{sample}/(^7Li/^6Li)_{standard} - 1] \times 1000$. The wide range of Li isotopic compositions in natural samples, between −15‰ and +45‰[31], provides a strong incentive for using $\delta^7Li$ values as a tool to ensure the traceability of lithium in LIB.

We first discuss the variability of Li isotope compositions between lithium deposits and in coexisting ores. The effects of extractive and refining metallurgy, cathode active materials synthesis and battery manufacturing on the intrinsic signatures of ores are then analysed and discussed. Finally, we discuss how Li isotope compositions can be used for ensuring the traceability and certification of lithium in LIBs.

## Results and discussion

The samples and analytical techniques we used are described hereafter under Methods. Figure 1 shows the samples analysed in this study with a known provenance. Supporting information is provided in Supplementary Figs. 1 and 2, Supplementary notes and the data are listed in Supplementary Tables 1 and 2.

**Isotope variability between lithium deposits and among coexisting ores.** Li isotope compositions of major deposits in China, Chile, Argentina, Bolivia and Australia were taken from previous studies[32–41] (Fig. 2). Their distribution in natural samples (spodumenes and brines) is shown in Supplementary Fig. 1.

For brines in South American salars in the "lithium triangle" of Bolivia (Salar de Uyuni), Argentina (Salar del Hombre Muerto, Salar de Olaroz, Salar de Ratones, Salar de Centenario, Salar de Pozuelos) and Chile (Salar de Atacama, Salar Grande, Salar de las Parinas, Salar de la Isla, Salar de Pedernales), the interquartile range (IQR) of Li isotope compositions is from +7.9 to +11.3‰ with a median value of +9.8‰ ($n = 103$)[35–40]. For brines of the Qaidam Basin in China, the IQR of Li isotope compositions is between +16.1 and +31.4‰ with a median value of +24.3‰ ($n = 20$)[41]. The origin of the lithium in brine is variously explained by low-temperature rock weathering, hydrothermal leaching, or magmatic origin with subsequent evaporation. In such deposits, dissolved lithium is commonly complexed with chloride as a LiCl species[42]. General theoretical considerations suggest that the lower coordination states and bond lengths should prefer the heavy isotope[43]. At ambient P-T conditions, the four-fold coordination $[Li(H_2O)_4]^+$ is the main cluster in aqueous fluids, whereas the coordination of Li in most solids is higher[31,44]. Li isotope fractionation in fluid–rock interactions, in particular rock weathering, results in preferential fractionation of the heaviest isotope ($^7Li$) into fluids with a magnitude inversely correlated to temperature[31,45]. This behaviour is consistent with low-temperature leaching experiments on tuff, yielding leachate that is +5‰ enriched in $^7Li$ relative to whole-rock Li[36]. Moreover, the Li isotope compositions of brines are also controlled by the incorporation of Li into secondary minerals, such as clays, removing the lightest isotope ($^6Li$) from the solution and enriching water in $^7Li$ (until + 10‰)[31]. Thus, the Li isotope compositions of brines result from the mixing of waters derived from various rock reservoirs[46] and from fluid–rock interactions at different temperatures. The enrichment of dissolved Li is consistent with literature data, which showed that $\delta^7Li$ values of South American brines (Bolivia, Argentina, Chile) (+7.9 to +11.3‰, $n = 103$, IQR) and China brines (+16.1 to +31.4‰, $n = 20$, IQR) are generally higher than upper continental crust (UCC) values (0 ± 4‰, 2σ)[31] (Fig. 2). While $\delta^7Li$ values of brines from the Qaidam Basin (China) are marked by strong enrichment (+16.1 to +31.4‰, $n = 20$, IQR) compared to South American "lithium triangle" brines (+7.9 to +11.3‰, $n = 103$, IQR), the variability of $\delta^7Li$ values in the latter brines is considerably greater than their differences (Fig. 2, Fig. S1).

For spodumenes in Australia (Yilgarn and Pilbara cratons) and China (West Kunlun), the IQR of Li isotope compositions is between −0.3 and +6.0‰, a median value of +2.8‰ ($n = 20$)[32–34]. The hard-rock deposits, mostly Li-rich granitic

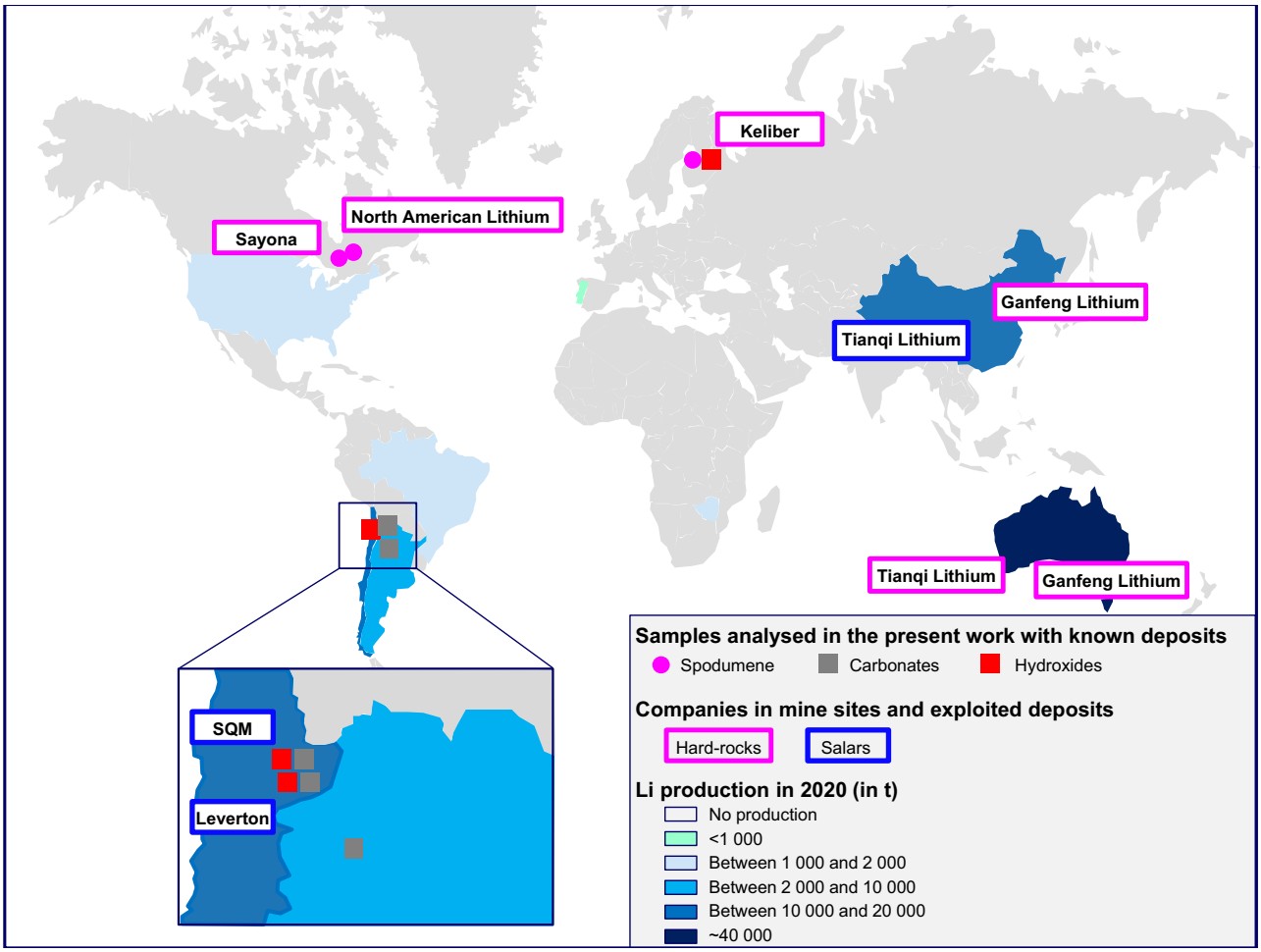

**Fig. 1 Map of world Li production in 2020 and location of lithium mining and refining companies studied in this work.** World mine production in 2020 is from USGS (2021)[10] data, except for the United States for which the value represented is the 2018 production[16] data. Spodumene concentrates, lithium hydroxides and carbonates analysed in this study with known deposits are shown, as is the lithium carbonate produced by Alfa Aeser in Argentina.

pegmatite, are interpreted as the product of fractional crystal-lisation from a parental granitic melt. Lithium is a major element in various minerals, such as amblygonite, bikitaite, eucryptite, lithiophilite, lithiophosphate, montebrasite, spodumene, and petalite in Li-rich granitic pegmatites. Among them, spodumene is the most exploited on a commercial scale[12]. Experiments and ab initio density functional perturbation theory (DFT) calcula-tions, showed that, during pegmatite crystallisation, $^6$Li prefer-entially occupies octahedral sites in spodumene, while $^7$Li favours tetrahedral sites in granitic melt[44,47]. The ab initio calculations by Liu et al.[48] predicted that Li isotope fractionation in Li-rich minerals has a notable linear correlation with the average Li-O bond lengths and Li coordination numbers; they demonstrated that the $\delta^7$Li values in minerals formed at the same crystallisation stage from a pegmatite melt in the order petalite>lithiopho-sphate>bikitaite>eucryptite>montebrasite>amblygonite>lithio-philite>spodumene. Therefore, in Li-rich granitic pegmatites, the $\delta^7$Li values of spodumene are lower than those of petalite. This isotope depletion in spodumene is confirmed by literature data, which showed that $\delta^7$Li values of spodumene in major deposits in Australia and China are, in contrast to salars, in the same order of magnitude ($-0.3$ to $+6.0‰$, $n = 20$, IQR) as UCC values ($0 \pm 4‰$, $2\sigma$)[31] (Fig. 2). While Li isotope compositions of spodumene from West Kunlun (China) are depleted in heavy isotopes ($-1.3$ to $+1.4‰$, $n = 8$, IQR) compared to Australian spodumene ($+3.8$ and $+9‰$, $n = 12$, IQR), the variability of $\delta^7$Li

values within Australian deposits (Yilgarn and Pilbara cratons) is more important than the differences between them.

The Li isotope composition of lithium deposits is linked to the physico-chemical conditions of ore-forming processes and varies within several tens of parts-per-thousand (Fig. 2). The different genesis of (supergene) salars and (magmatic) hard-rock deposits explains why $\delta^7$Li values of brines are generally higher ($+7.9$ to $+11.3‰$, $n = 103$, IQR and $+16.1$ to $+31.4‰$, $n = 20$, IQR) than those of spodumene deposits ($-0.3$ to $6.0‰$, $n = 20$, IQR). This variation in $\delta^7$Li values could discriminate a salar from a spodumene origin (see discussion below), but also between deposits of the same type (Australia versus China for hard-rock, South America versus China for salars).

**Effects of extractive and refining metallurgy, cathode active materials synthesis, and battery-cell manufacturing.** The con-centrating, extracting, or processing of the lithium contained in ore deposits will affect their $\delta^7$Li value. In particular industrial processes, involving chemical transformation with kinetic isotope effects and low-recovery-yield/high-lithium-loss, can induce sig-nificant isotope fractionation between industrial and natural samples. The LIB production chain includes several industrial processes: (i) The hard-rock extractive metallurgy process starts with producing a spodumene concentrate, increasing the lithium content by separating undesirable minerals from ore through

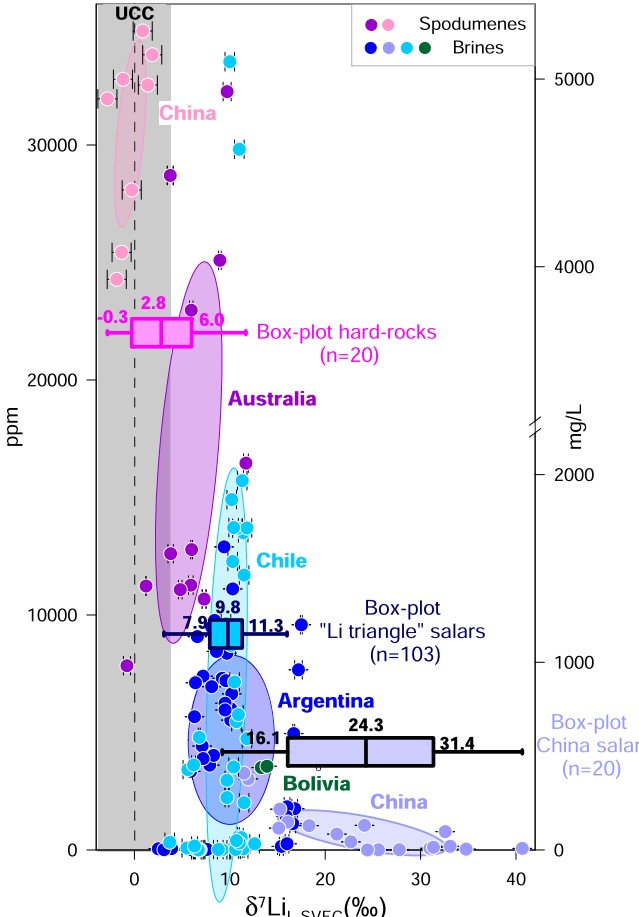

**Fig. 2 δ⁷Li value versus lithium content (ppm, mg/L) for various lithium deposits.** Spodumenes from West Kunlun (China) and the Yilgarn and Pilbara cratons (Australia)[32–34]; and brines from the Qaidam Basin (China), Salar del Hombre Muerto, Salar de Olaroz, Salar de Pozuelos (Argentina), Salar de Atacama, Salar Grande, Salar de las Parinas, Salar de la Isla, Salar de Pedernales (Chile), and Salar de Uyuni (Bolivia)[35-41]. Error bars represent the 2σ associated with δ⁷Li values. The isotopic compositions of different Li deposits are shown by probability ellipses (confidence level $p = 0.68$). Data are shown with blue box-plots for "Li triangle" salars ($n = 103$), China salar ($n = 20$) and pink box-plots for hard-rocks ($n = 20$). The vertical grey line labelled "UCC" represents upper continental crust values (UCC): 0 ± 4‰ (2σ)[31].

physical separation (comminution, flotation and magnetic separation)[49]. The concentrate is then calcined at >1000 °C, causing the restructuring of α-spodumene to β-spodumene which is readily dissolved in acid[50]. The traditional sulphuric acid process was the first to efficiently extract lithium from spodumene in the 1950s (85–90% lithium yield at the time) and was scaled-up shortly after (yield over 90%)[11,12]. In this process, the roasted β-spodumene is leached with sulphuric acid and mixed with sodium carbonate to precipitate lithium carbonate. A last step, adding calcium hydroxide, can be used for obtaining lithium hydroxide from lithium carbonate[50]. The material used as isotope standard, Li carbonate L-SVEC, was purchased from Lithium Corporation of America (or American Lithium)[30], prepared from Li ore (mostly spodumene) from Foote Mine (Kings Mountain, North Carolina, USA) using traditional extraction with sulphuric acid leaching https://www.americanlithiumcorp.com. Grégoire et al.[51] showed that the Foote Mine ore and the derived carbonate have a similar Li isotope signature taking into account analytical uncertainty, indicating that the sulphuric acid process does not

cause Li isotope fractionation. As an alternative to the traditional process discussed above, Outotec and Keliber of Finland announced in early 2019 a new process, totally sulphate and acid-free, for producing lithium hydroxide directly from calcined β-spodumene[52]. After calcination, two-stage alkaline leaching (pressure- and conversion leaching) produces a hydroxide solution and analcime ($NaAlSi_2O_6.H_2O$). The overall lithium-leaching extraction yield from concentrate is 84–94%[52]. The typical lithium-processing impurities Fe, Al, Ca, Mg and P are then removed from the solution by cation-exchange resins with iminodiacetate or aminophosphonate (ion-exchange purification). Finally, $LiOH·H_2O$ is solidified by pre-concentration and vacuum crystallisation. Figure 3 shows the spodumene concentrate, β-spodumene, analcime and Li hydroxide samples provided by Keliber. In contrast with the American Lithium product, the Finnish samples show strong fractionation between spodumene concentrate and the produced Li hydroxide ($Δ^7Li_{hydroxide-spodumene \ concentrate} = +5.5‰$). Calcination does not cause isotopic fractionation, as spodumene concentrate and β-spodumene have the same lithium isotopic signature (Fig. 3). Concerning the leaching step, we can estimate the Li composition of the product with a Rayleigh model (Supplementary Fig. 2). Using the starting composition of δ⁷Li in ores (+1.1‰), the δ⁷Li of the analcime by-product (−0.9‰) and the lithium-leaching extraction yield given by Keliber (84% to 94%), the estimated δ⁷Li values are between +1.3 and +1.5 ‰ for Li in solution (Fig. 3). This shows that leaching does not lead to significant Li isotope fractionation ($Δ^7Li_{Li+-β-concentrate} + 0.2$ to +0.4‰). Concerning ion-exchange purification, strong Li isotope fractionation occurs during ion-exchange chromatography[53]. The heavy ⁷Li isotope passes more rapidly through the exchange resin than ⁶Li, requiring a 100% yield to avoid isotopic fractionation in the eluent during Li chemical preparation[31] (see Methods, hereafter). We carried out laboratory experiments for estimating the fractionation factor between Li+ and purified Li+ (eluent) due to purification by cation-exchange resins (see Supplementary note for more details). These experiments showed that even a 95% yield causes strong fractionation between Li+ and purified Li+ ($Δ^7Li_{ purified \ Li+-Li+} > +8‰$). Concerning the crystallisation process, due to no change in the coordination number between Li in aqueous solution and Li hydroxide monohydrate (both tetrahedrally coordinated sites)[54], this is not expected to result in significant Li isotope fractionation.

(ii) The salar extractive metallurgy, i.e. lithium production from brine, depends on its composition, volume and accessibility, as well as on its amenability to local processing[49]. At Salar de Atacama (Chile) and Salar de Olaroz (Argentina), the processing flow sheet used by the Rockwood, SQM and Orocobre companies is referred to as the 'Silver Peak' method, where it was first developed in Nevada (USA) by Foote Mineral in 1960s[50]. Brines are pumped to the surface to be concentrated by solar evaporation in ponds. This concentration causes precipitation of sodium, potassium and magnesium chlorides[50]. In addition, at Salar del Hombre Muerto (Argentina), lithium brine is first concentrated by ion absorption onto polycrystalline alumina before solar evaporation[50]. However, heavy Li loss is observed due to evaporated brine caught in salts precipitated during evaporation, and the maximum recovery of Li from evaporation is ~80%[55]. Concentrated brine is then transferred to processing facilities where reagents are added to remove impurities and to produce lithium compounds via precipitation/crystallisation[50]. In the SQM process, the Salar de Atacama brine is drained from the evaporation ponds once lithium concentration in the brine reaches ~6% Li, or the saturation point of lithium chloride, and transported to the Salar del Carmen plant via truck[50]. The brines undergo solvent extraction to remove boron, and soda ash is added to precipitate and filter out magnesium carbonate. Then,

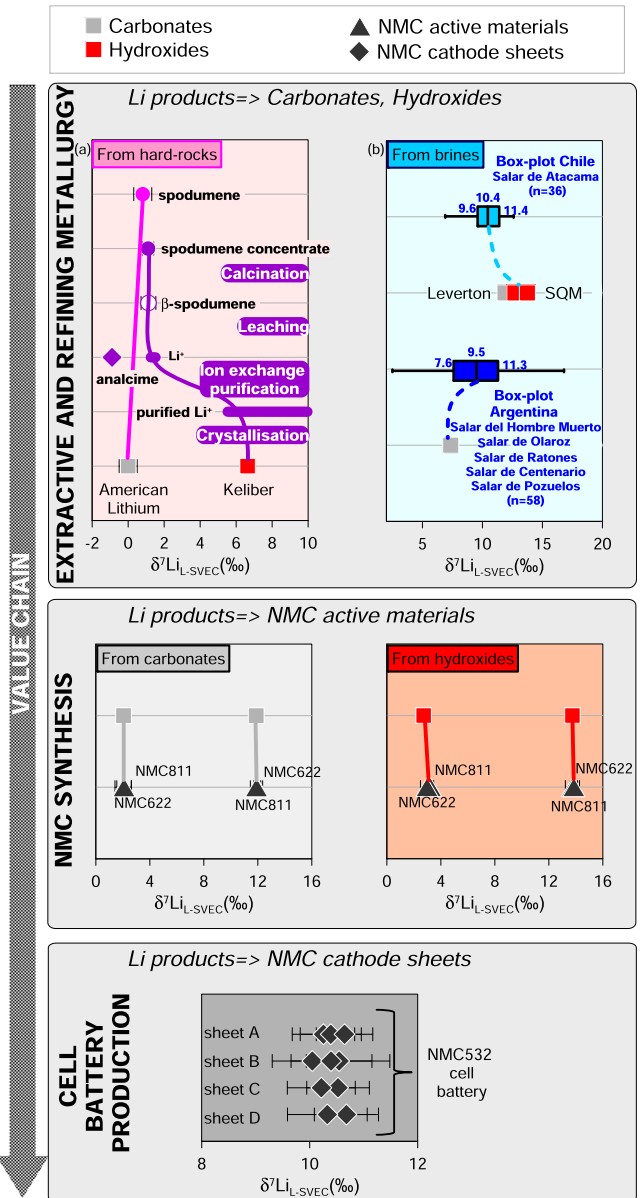

**Fig. 3 Lithium isotope fractionation in various products containing Li (carbonates, hydroxides, NMC active materials, NMC cathode sheets) manufactured during the LIB production chain.** Extractive and refining metallurgy (**a**) for hard-rock-based lithium sources, with spodumene and Li carbonate from American Lithium, and spodumene concentrate, β-spodumene, and Li hydroxide from Keliber (this study). $\delta^7Li$ values for $Li^+$ ($\alpha = 1.0007$, 1.0011; $R = 84$–94%) and purified $Li^+$ ($\alpha = 1.055$; $R = 95$–98%) are estimated as explained in the main text. **b** Brine-based lithium sources: Li hydroxide and -carbonate from Leverton, SQM and Alfa Aesar ($n = 5$, this study). Blue box-plots are Argentinian salars ($n = 58$)[35,36,38,39] and Chilean salar de Atacama ($n = 36$)[37,39,40] data. NMC active materials synthesis with NMC622 ($n = 2$) and NMC811 ($n = 2$) produced in this study from lithium carbonate (Li13, Li18) and -hydroxide (Li01, Li17). Cell battery production with various pieces of cathode sheets ($n = 8$) coming from the same NMC532 cell battery (this study). Error bars represent the $2\sigma$ associated with $\delta^7Li$ values.

the concentrated brine is heated and reacted with additional soda ash to precipitate lithium carbonate, which is filtered, washed and dried in a rotary drier[50]. As a final step, the Li carbonate can be converted to Li hydroxide by adding calcium hydroxide. For the European market, this last stage takes place at processing plants

in Russia[50]. The process used by Leverton on Salar de Atacama brines is not described in the literature and Leverton does not disclose its processing information. However, the $\delta^7Li$ values of the carbonate and hydroxide produced by SQM and Leverton from Salar de Atacama brines are close to each other ($\Delta^7Li_{SQM-Leverton} + 1.0$ to $+1.4$‰), indicating that the metallurgical processes used can be similar. The $\delta^7Li$ value of the carbonate produced in Argentina ($+7.4$‰) is slightly lower than literature data for brines from Salar del Hombre Muerto, Salar de Olaroz, Salar de Ratones and Salar de Pozuelos ($+7.6$ to $+11.3$‰, $n = 58$, IQR). The $\delta^7Li$ values of the Li carbonate ($+11.9$ to $+13.3$‰) and hydroxide ($+12.7$ to $+13.7$‰) produced by Leverton and SQM from Salar de Atacama brines are slightly higher than those of the Salar de Atacama brines determined in previous studies ($+9.6$ to $+11.4$‰, $n = 36$, IQR) (Fig. 3). This difference between natural samples and products may be either due to the fact that the literature data are not representative of brines exploited by salt producers, or to the fact that isotopic fractionation occurs during extraction, in particular during evaporation when Li losses are highest. This point shows the need for working in collaboration with salt producers to evaluate the significance of their products. The conversion of Li carbonate to Li hydroxide does not induce isotopic fractionation since the $\delta^7Li$ values for these products are close for SQM and Leverton samples: $\Delta^7Li_{hydroxide-carbonate} + 0.4$ to $+0.8$‰ (Fig. 3). Please note that the $\delta^7Li$ values are also close for the Li hydroxide and -carbonate produced by Tianqi Lithium (China) (Fig. 4), reinforcing the hypothesis that the conversion from carbonate to hydroxide does not alter the Li isotope signature.

(iii) The cathode active material synthesis. Producing batteries with a high energy density requires active materials with a high volume density. Coprecipitation synthesis is commonly used for producing dense lithium-layered oxide materials with spherical particles[56]. In such a synthesis, a mix of nickel, cobalt, and manganese sulphates in appropriate amounts for producing the targeted NMC, is dissolved in water. This sulphate solution and an ammonium hydroxide solution are pumped together into a stirring tank reactor, with the addition of a sodium hydroxide solution for maintaining the reaction at basic pH. After an ageing period, the resulting precipitate is recovered by filtration. This first synthesis step leads to a mixed-metal hydroxide, which is then mixed with a lithium salt. The resulting powder is calcined at a high temperature to produce the active material (see Methods, hereafter, for more details on NMC622 and NMC811 synthesis). Figure 3 shows the $\delta^7Li$ values for active materials (NMC622 and NMC811) synthesised from Li carbonate and hydroxide (Li01, Li13, Li17, Li18) for this study. Regardless of the type of NMC produced (NMC622 or NMC811) and the precursor used (Li hydroxide or Li carbonate), the $\delta^7Li$ values of the precursor and the product are similar considering analytical uncertainty. Synthesis of active material does not induce significant isotopic fractionation between the lithium salt and the active material.

(iv) The battery-cell manufacturing. A cathode sheet consists of a current collector, typically aluminium foil, on which a fine powder of active material with polyvinylidene difluoride (PVDF) and carbon black is deposited on two sides. The battery-cell assembly consists of alternating anode-, separator- and cathode sheets in a cell pack, filled with electrolyte. As these steps do not involve any chemical transformation of the lithium contained in the active material, they cannot cause any significant isotopic fractionation between the active material and cathode sheet. The Li isotope compositions of different sheets of the same battery cell, whether covered or not by electrolyte, are similar when considering analytical uncertainty (Fig. 3). Such homogeneous composition indicates that a battery can be characterised by a single $\delta^7Li$ value determined by a punctual analysis on a single

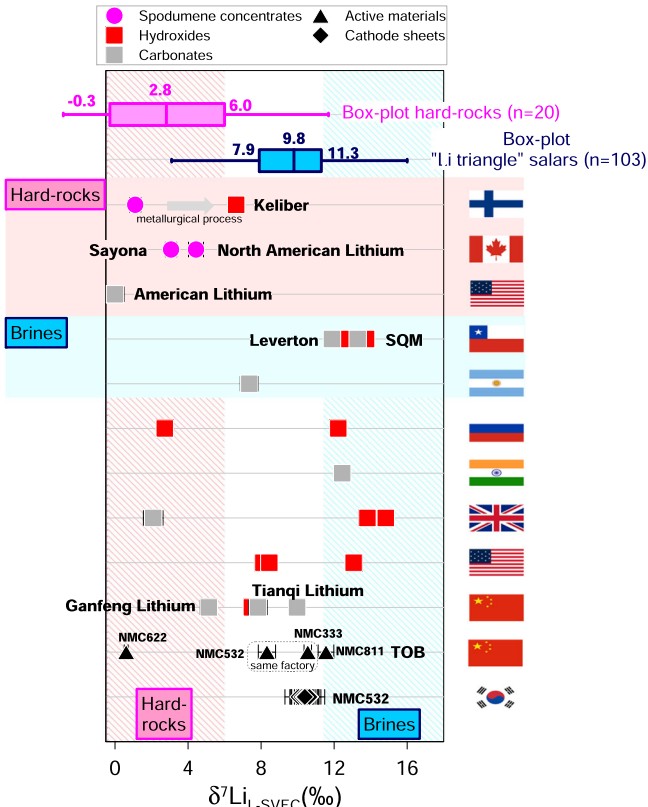

**Fig. 4 Lithium isotope compositions of various battery precursors and components produced around the world (Finland, Canada, USA, Chile, Argentina, Russia, India, UK, China, and South Korea).** Spodumene concentrates ($n = 3$), lithium hydroxides ($n = 11$), lithium carbonates ($n = 8$), cathode active materials ($n = 5$), cathode sheets ($n = 8$). The lithium carbonate produced by American Lithium (L-SVEC) is also shown[51]. Error bars represent the 2σ associated with δ7Li values. Data are shown with blue box-plots for "Li triangle" salars ($n = 103$)[35–40] and pink box-plots for hard-rocks ($n = 20$)[32–34]. The "hard-rocks domain" and "brines domain" defined in this study are shown by pink and blue hatched areas, respectively. The "unknown origin domain" is shown by the white area between the two hatched areas.

sheet. The δ7Li value of this battery ($+10.4 \pm 0.4‰$, 2σ) is in the same value range as Korean LIBs ($+8.5‰$, $+2.4‰$, $+3.1‰$, $+12.6‰$) analysed by a previous study evaluating the impact of anthropogenic input on lithium content in the environment[57].

In conclusion, other than the sulphuric acid process, the extraction and purification processes discussed above tend to increase the δ7Li value of the produced salt compared to its initial/natural Li isotope signature. The last, supplementary, step of the lithium transformation chain (conversion of Li carbonate to Li hydroxide) does not introduce isotope fractionation. The other stages of battery manufacture (cathode active material synthesis, battery-cell manufacturing) neither induce significant isotopic fractionation between the lithium salts and the end product, which has a homogenous Li isotope composition.

**Assessing the geochemical traceability of lithium**. Geochemical traceability is used to try and answer the question "*What is the origin of unknown lithium?*", determining the origin (mine site, refining plant) of a material (ore, product) using measurable and quantifiable material properties. For that, materials must have measurable compositions/properties that differ depending on their geological genesis or manufacturing. The Li isotope compositions of lithium deposits are related to the physicochemical

conditions of ore-forming processes; differences in their genesis lead to higher δ7Li values for brines ($+7.9$ to $+11.3‰$ and $+16.1$ to $+31.4‰$) than for hard-rock deposits ($-0.3$ to $+6.0‰$). However, extraction and purification processes, other than the traditional sulphuric acid process, tend to modify the initial/natural signature by increasing the δ7Li values by up to $+5.5‰$. Though such process-related fractionation tends to erase the link of a sample to its geological origin, it can also serve to differentiate lithium salts produced from ores of similar origin, but for which the extraction process may have a different environmental or social impact. For example, this fractionation could discriminate lithium salts produced from spodumene using the traditional sulphuric acid process or using an alternative process without sulphuric acid, such as the Outotec and Keliber process.

Despite the uncertainty related to process-related isotopic enrichment and the lack of data on deposits, we can establish a first estimate of ranges of Li isotopic values for which the probability of the Li salt belonging to either hard-rock-based or brine-based lithium sources is high. This first estimation will be refined as more data are acquired on the different deposits and the various extraction processes. For δ7Li values below $+6‰$ (the third quartile of hard-rock data), the probability is high that the sample was obtained from hard-rock, whereas δ7Li values over $+11.3‰$ (the third quartile of "Li triangle" salars data, the values for the Chinese salar being even higher) indicate a sample probably obtained from brine. However, samples with δ7Li values value between $+6‰$ and $+11.3‰$ fall in the "unknown origin domain". Considering samples of known deposits (Fig. 1), the three spodumene concentrates (North American Lithium, Sayona and Keliber) are within the "hard-rock domain" and the four Li salts from Atacama salar brines (Leverton and SQM) fall within the "salar domain" (Fig. 4). The Li hydroxide from Keliber and the Li carbonate from Argentina fall in the "unknown origin domain". For the other salts, for which only the country of the last refining stage is known, there is a heterogeneity of the Li origin within the same country. For example, samples produced in Russia and the UK come from salars and hard-rock (Fig. 4). The Li carbonate of Ganfeng Lithium, which produces Li salts from Australian and Chinese spodumene concentrates, is in the "hard-rock domain", while the products of Tianqi Lithium that has a more diversified supply (salar or spodumene), are in the "unknown origin domain".

As we saw that the synthesis of active material and the manufacturing of battery cells do not induce significant isotopic fractionation, the ranges of Li isotopic values established above can be used as a first estimate for determining the origin of lithium in active materials and battery cathode sheets. The δ7Li values for active materials produced by TOB (China) are variable, including for materials produced in the same factory (NMC532 and NMC333) (Fig. 4). Except for the TOB active materials NMC622 (in the "hard-rock domain") and NMC811 (in the "salar domain"), the other TOB samples and cathode sheets from the Korean battery maker fall in the "unknown origin domain". These results show that the supply of lithium to the battery industry is based on economic criteria, with no preference for hard-rock- or brine-based lithium.

Though these results show that identifying the origin of an unknown lithium product is a challenging issue, the large diversity of Li isotopic signatures for secondary products demonstrates that δ7Li values, like a fingerprint, can be a useful tool for certifying the origin of lithium in LIB.

**Towards a methodological approach for certifying a responsible and sustainable lithium supply chain**. Our analytical method, based on lithium isotope fingerprints, can help

controlling and certifying the origin and trade of lithium production. It is an independent, reliable and tamper-proof approach to auditing the document-based traceability system sought after by end-users (carmakers, consumer electronics companies, etc.), by answering the question: "*Does the lithium correspond to its declared origin?*". Traded materials can be analysed to provide additional credibility to document-based traceability systems with due-diligence concepts for raw material supply chains. Implementation of a certification system for lithium will boost the development of a responsible, sustainable and stable supply of raw materials for batteries, guaranteeing the respect and protection of human rights and the conservation of the environment along the value chain. The development of lithium certification is of critical importance, especially in the context of the political will to re-industrialise battery production in Europe or in the US, which are defending sustainable battery manufacturing projects. Such certification would be in accordance with the recent EU regulation for responsible and sustainable sourcing of several other raw materials, such as tin, tantalum, tungsten and gold, and the consumers' interest in sustainable products. The principle of this analytical method is the same as that used for the traceability of gold and coltan[28,29], which verifies whether the product corresponds to its declared origin by comparing the sample in question with reference samples of known origin stored in a database. As the Li isotopic signature is conserved from lithium salt to the battery, it is possible to develop this control along the value chain.

Adopting such an approach will require guidelines for collecting reliable data on sample provenance, and for a reference database with comprehensive and up-to-date data on Li products available on the market. To this end, reference samples must be collected of raw and processed materials from locations worked by one or several companies for a certain period of time, such as a year. In particular, it must be verified whether samples produced by the same company from the same deposit are more closely related to each other than samples produced by another company from a different deposit. This approach is only possible if robust data on within-deposit variations are available; moreover, the database must be active, as new orebodies of deposit are exploited, new extraction sites are open, and new mining/refining companies enter the market. The limitation of this approach will be the overlaps in the data of Li products from different locations or salt producers. A specific statistical data evaluation strategy is needed for evaluating matches between unknown and reference samples from mine sites or processing plants declared as the origin of the unknown sample.

Beyond this study, further challenges for developing lithium certification will consist in enlarging the database and assessing the applicability of this approach to non-conventional Li sources (e.g., geothermal waters, clay minerals) to support the future development of the global lithium supply chain.

## Methods

**Sample description.** Three spodumene concentrates from mining companies in Finland (Keliber Oy) and Canada (North American Lithium, Sayona Québec) were sampled. Keliber also provided processed products: β-spodumene, analcime ($NaAlSi_2O_6 \cdot H_2O$) and lithium hydroxide monohydrate (LiOH. $H_2O$).

Keliber Oy (Keliber) operates spodumene deposits located in Central Ostrobothnia province (Finland)[50], and produces battery-grade lithium hydroxide in its chemical plant https://www.keliber.fi/en/.

North American Lithium operates an open-pit mine in La Corne (Abitibi, Québec, Canada) and plans the opening of a lithium carbonate plant http://na-lithium.com/.

Sayona Québec (Sayona) is a subsidiary of Sayona Mining, an emerging lithium miner with projects in Québec and Western Australia. It further owns the Authier Lithium Project in Québec for the development of an open-pit spodumene mine https://www.sayonaquebec.com/.

We also analysed eight samples of lithium carbonate ($Li_2CO_3$) and ten samples of lithium hydroxide monohydrate (LiOH. $H_2O$) of battery-grade purity (Li > 99.5%) coming from various chemical companies (Alfa Aeser, Acros Organics, Fluka, Sigma Aldrich, Fisher Chemical, Leverton), and from mining/refining companies that manufacture cathode active material. In particular, we analysed Li salts from three of the world's top five producers of lithium chemicals (SQM, Ganfeng, and Tianqi)[8]. We assumed that the lithium carbonate produced by Alfa Aeser in Argentina (Li 11) was made from Argentinian salars.

Leverton-Clarke (Leverton) operates a processing plant in Basingstoke, Hampshire, UK https://www.levertonlithium.com/; they produce lithium hydroxide and battery-grade carbonate from Salar de Atacama brine (personal communication).

Sociedad Quimica y Minera (SQM) extracts lithium brine from the Salar de Atacama in northern Chile. It is the world's largest producer of lithium carbonate and a major producer of lithium hydroxide. SQM operates a lithium carbonate and -hydroxide plant at the Salar del Carmen facilities at La Negra, near Antofagasta[50]. Lithium carbonate, supplied by SQM, is also transformed at processing facilities in Russia to lithium hydroxide, which is redistributed mainly in the European market[50].

Jiangxi Ganfeng Lithium (Ganfeng Lithium) operates a spodumene mine in China (Ningdu) and has a 50% equity ownership in the Mt. Marion lithium mine in Western Australia[50], as well as exclusive supply agreements with Pilbara Minerals (Pilgangoora and Altura projects) in Australia http://www.ganfenglithium.com/about3_en.html. Ganfeng Lithium operates a number of subsidiaries, undertaking lithium exploration in Ireland, Canada, Australia, Mexico and Argentina, lithium processing in China, and marketing of lithium products in the Chinese and international markets[50].

Sichuan Tianqi Lithium Industries (Tianqi Lithium) is a state-owned Chinese enterprise operating multiple lithium operations and projects, mainly in China and Australia. It holds a 51% share in the Greenbushes Mine of Western Australia[50], is the largest producer of lithium mineral concentrates, and exploits the brines of Zhabuye Salt Lake on the Tibetan Plateau (China)[58]. Two Li-processing plants are operated by subsidiaries of Tianqi Lithium in China, in Sichuan and Jiangsu provinces. They produce lithium chemicals from Li products imported from a diversified supply base (salar or spodumene origins)[50].

Xiamen TOB New Energy Technology (TOB) is a Chinese company specialised in lithium-ion battery research and manufacturing. It provides equipment, materials and comprehensive battery production-line solutions for international companies and research institutions (BMW, Daimler-Benz, A123, SKC, MIT, IIT, etc.) https://www.tobmachine.com/company_d1, and produces cathode active materials, four of which (NMC333, NMC532, NMC622 and NMC811) were sampled. Samples NMC333 and NMC532 were produced in factory A, whereas NMC622 and NMC811 were each produced in two other factories (B and C).

For this study, we synthesised two types of active materials (NMC622 and NMC811) from lithium carbonate (Li13, Li18) and -hydroxide (Li01, Li17) at CEA LITEN (Commissariat à l'Energie Atomique et aux énergies alternatives Laboratoire d'Innovation pour les Technologies des Energies nouvelles et les Nanomatériaux). Layered lithium-oxide material was synthesised by coprecipitation using commercial sulphate reactants from Sigma Aldrich. In a standard synthesis, three different solutions, containing all reactants, were prepared. The transition metal-ion solution was obtained by dissolving $NiSO_4 \cdot 6H_2O$ (127.4 g for NMC622, 169.9 g for NMC811), $MnSO_4 \cdot H_2O$ (27.3 g for NMC622, 13.7 g for NMC811) and $CoSO_4 \cdot 7H_2O$ (45.4 g for NMC622, 22.7 g for NMC811) in 400 g water. The ammonium hydroxide solution was produced by mixing 150 g $NH_4OH$ (28% from Sigma Aldrich) in 233 g water, and the sodium hydroxide solution resulted from dissolving 81.6 g NaOH (from Sigma Aldrich) in 400 g water. The transition metal-ion solution and the ammonium hydroxide solution were pumped directly into the reactor, the pH being kept at 11 during synthesis through controlled injection of the hydroxide solution. After the introduction of the reactive solutions, the mixture was aged for 3 hours in the reactor, before recovery by filtration of nickel-manganese-cobalt hydroxides [$Ni_{0.6}Mn_{0.2}Co_{0.2}(OH)_2$ or $Ni_{0.8}Mn_{0.1}Co_{0.1}(OH)_2$]. The product was washed several times with hot water in order to remove residual sodium and sulphate species, and finally, the hydroxide was dried overnight in an oven at 80 °C. To obtain the final NMC material, the hydroxide was intimately mixed with excess (3.3%) lithium salt and the mixture was fired at 850 °C for 24 h under air for producing NMC622, and at 925 °C for 12 h under oxygen for producing NMC811.

A large prismatic "automotive-grade" battery cell (30 × 9 cm) from South Korea with an NMC532 cathode, containing 52 cathode sheets and 53 anode sheets, was sampled as well.

**Reagents and materials.** All plastic and Teflon equipment for this study was acid-cleaned before use. All acids were purified by sub-boiling distillation before use. The water was distilled "Milli-Q" water with a resistivity of 18.2 MΩ cm (Millipore®). Cation-exchange resin AG 50 W − X12 (200–400 mesh) and hydrogen from BioRad® were used for Li purification.

**Sample preparation.** The "automotive-grade" battery cell was opened in the EDF-LME (*Électricité de France-Laboratoire des Matériels Electriques*) R&D laboratory, after complete discharge for safety reasons. Four cathode sheets (A, B, C, D) were selected to provide representative samples of the cell. Sheets A and B were rinsed

with "Milli-Q" to remove any electrolyte residues, whereas sheets C and D were left untouched.

The samples were then further prepared in the BRGM (*Bureau de Recherches Géologiques et Minières*) laboratory. Several 3-cm-wide strips were cut from each sheet at different places with a ceramic chisel, and the front (A1, B1, C1, D1, A7, B3) or rear (A8, B6, C6, D6) faces were carefully scratched off with a ceramic lancet to avoid damaging the collector, composed of aluminium foil. About 200 mg of cathode active materials were calcined at 550 °C and dissolved in concentrated acids ($HNO_3$, $HClO_4$, HF, HCl) on a hot plate in the cleanroom. About 200 mg of spodumene concentrate and analcime were dissolved in concentrated acid using the same protocol. After drying, the residue was diluted in 0.5 M $HNO_3$. About 200 mg of lithium carbonate and -hydroxide were also dissolved in 0.5 M $HNO_3$.

**Lithium isotope analysis**. Li concentrations were measured using an X Series II ICP-MS (Thermo Fisher Scientific) in the BRGM laboratory. A sample volume of ~100 ng Li was dried on a hot plate in the cleanroom. For cathode active materials, the residue was dissolved in a mixture of 0.2 M HCl. Lithium was separated from matrix elements using an AG 50 W − X12 resin (200–400 mesh)[59], before drying and re-dissolving in 0.5 M $HNO_3$. To avoid isotope fractionation of Li due to chemical purification, the Li recovery from this protocol was checked by analysing one aliquot before and after chemical separation by ICP-MS: recoveries were consistently close to 100%. The other samples were directly dried and re-dissolved in 0.5 M $HNO_3$. Total procedural blanks were measured to verify the cleaning procedure; such blanks are generally less than 30 pg, representing >0.03% of the lithium mass analysed.

Lithium isotope compositions were measured at a concentration of 50 µg/L with a Thermo Fisher Scientific Neptune MC-ICP-MS—upgraded to 'Neptune Plus'—in the BRGM laboratory, following the procedure developed before[59]. The Li-isotope composition of each sample was expressed in δ-notation relative to the mean value of the bracketing Li standard (L-SVEC): $\delta^7Li = [(^7Li/^6Li)_{sample}/(^7Li/^6Li)_{standard} − 1] \times 1000$. The quality of Li-isotope analyses was controlled by regular measurements of "in-house" standards, whose long-term reproducibility is 0.5‰ (2σ). The external reproducibility (2σ) reported in the various figures and tables was typically ±0.4‰, calculated by measuring the same sample multiple times over the many analytical sessions.

## Data availability

All data generated or analysed during this study are included in the Supplementary Information.

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

## Acknowledgements

This work was financially supported by BRGM (TRAÇABILITÉ BATTERIES, PEX BATTERIES), and EDF Lab (TREE and LME Departments, PEM Project). Some state-of-the-art elements come from the SURFER project, financed by the French Environment & Energy ministry (ADEME, grant number 1605C0025). The authors wish to thank companies (Keliber, SQM, North American Lithium, Sayona Québec, Leverton) and universities (UQAT, Tianjin University) for providing samples. Thanks are also due to the batteries-traceability team project (BRGM and EDF Lab) for helpful discussions. Special thanks to colleagues from BRGM's Water, Environment, Analysis and Processes Division. H.M. Kluijver edited the final English version of this paper.

## Author contributions

A.M.D., D.M.C., G.L. and A.C.-T. designed the research. S.P., C.G. and A.U. conducted the samples preparation and chemical analyses. D.P. conducted the cathode active materials synthesis. All authors contributed to data interpretation and preparation of the final manuscript.

## Competing interests

The authors declare no competing interests.
