## [Peer Review File · Nature Communications]

Tracing the origin of lithium in Li-ion batteries using lithium isotopesREVIEWER COMMENTS

Reviewer #1 (Remarks to the Author):

The paper is a very important contribution to the ongoing discussion on applicability of geochemical fingerprinting technologies to raw material supply chains. It is well written, well illustrated (except for Figure 4 which is too colourful and confusing to read- probably remove the flags on the right side), and clearly deserves to be published. The weakness is the limited data set for primary Li deposits, the strength is the approach, including an experimental approach, taken to investigate Li isotope fractionation. In the following, I will give some thoughts on the study and its potential applicability.

The study "Tracing the origin of lithium in Li-ion batteries using Li isotopes" describes a fingerprinting technology and concept for Li batteries. So far, it was not possible to receive information on the origin of lithium used in batteries. In the light of efforts to reach transparent and responsible supply chains for materials, this is a highly valuable first approach. It is especially important for those materials that are part of the "Green Deal", where monitoring of CO₂ footprints for all process steps is essential; thus, the origin of all materials used in battery production must be monitored, from mining through processing into the battery factory and into the production of cars.

The authors use Li isotopes as the only tracing parameter. This has one major reason: processing Li ores and brines into intermediate materials used in battery production will likely destroy other tracing parameters such as trace element compositions. These are frequently used in fingerprinting studies for raw materials but usually do not "survive" the processing steps and are no longer indicative parameters once they leave the metallurgical processes. Therefore, it is a good approach to focus on the stable isotope composition of the major element in the mined ore (in this case, Li) and to investigate if this composition remains unaffected during any processing steps. This was done in the present study. The authors use two types of raw materials that, from a geological point of view, must reveal large differences in Li isotope composition. These differences have already been known from previous studies. Due to the limited number of Li isotope analyses from hard rock pegmatite deposits, which are one major supplier of Li via Li silicate (mostly spodumene), no major difference between such primary deposits could be found. However, when looking at the literature data on their Fig. S1, there is a wide spread that likely will discriminate different Li pegmatite deposits once more material for a better statistical evaluation has been analysed.

The authors show on Fig. 2 that the interquartile range of $\delta^7\text{Li}$ in Li pegmatites varies from -0.3 to +6.0 permil, with individual analyses ranging from -3 to +12 permil (Fig. 1). $\delta^7\text{Li}$ of 1.1 to 4.4 permil in three spodumene concentrates analysed in the present study is in the expected range.

For the second important source of Li, brine deposits in South America and China, the authors found a $\delta^7\text{Li}$ interquartile range of 7.2 to 11.9 permil in literature data, with individual data ranging from 2 up to 40 permil. The number of data points, especially for China, is not enough to calculate statistically valid numbers, resulting in high interquartile ranges. Nevertheless, evaluation of the literature data and the limited number of own data suggests that Li derived from pegmatites and salars can be discriminated using Li isotopes. But it must be kept in mind, that there is a significant overlap between pegmatite and salar Li that might render interpretation of $\delta^7\text{Li}$ values in the +2 to +12 permil range difficult because of possible overlap. Thus it is of utmost importance to develop a reliable set of data for each individual deposit from which Li is processed. Evaluation of the Li source is only possible if robust data on within-deposit variations are available; for pegmatite deposits, this should comprise repeated analyses of concentrates delivered to the plant for a certain period of time, e.g. within a year. It must also be kept in mind that Li isotope compositions may vary if further orebodies or other parts of – commonly zoned – pegmatites are being put in operation. In the case of Li from brines, repeated analyses of products must also be carried out; the large variation in Argentinean and Chinese deposits shown by literature data suggests inhomogeneity of Li isotopes within salars. The reason for this must be clarified.

In the present study, the authors put much emphasis on proving that Li isotope composition remains stable during purification. For pegmatites, calcination and sulphuric acid processing do not cause Li isotope fractionation. The new process developed by one company producing purified Li by ion exchange purification, however, leads to significant fractionation towards heavier isotope

composition.

For Li produced from salars in Argentina, $\delta 7\text{Li}$ in Li carbonate is similar to the literature data from brines. Heavier isotope compositions measured in carbonate and hydroxide products from two companies using Salar de Atacama brines are explained by inadequate literature data, or by fractionation during extraction. This important point should be more carefully studied in the future. Finally, cathode material synthesized from Li carbonate and hydroxide was analysed and it was shown that Li isotopes did not fractionate during synthesis.

The conclusions of the study are complex. On the one hand it can be shown that traditional processing of pegmatite-Li leaves light Li isotope compositions typical of the primary minerals and thus would make identification of Li origin in batteries possible. On the other hand the alternative extraction and purification process for pegmatite-Li leads to significantly heavier Li isotope composition, similar to brine-Li. The reason for unusually heavy compositions of Li salts produced from Atacama brines is not clear, but such heavy signatures are clearly not expected from pegmatite-Li. Therefore the authors tentatively split their data into three groups claiming that $\delta 7\text{Li} < 6$ permil most likely are derived from hard-rock pegmatite deposits, and values > 11.9 permil indicate a brine origin. All data in between fall into the "unknown origin domain". These three categories provide a good starting point for the setup of a Li traceability process in the battery supply chain. However, there are obvious weaknesses mainly due to inadequate sampling. If the data base will be improved in a way that robust data on Li isotope chemistry within the deposits currently producing Li are available, and final questions regarding fractionation during processing are resolved, the method has a high potential to be applicable in the future.

Reviewer #2 (Remarks to the Author):

General comments

This manuscript focuses on the utility of Li isotope measurement in tracing sources of lithium used in Li ion batteries. The authors point out that there is a growing demand for Li for use in such batteries and ensuring that both socially and environmentally responsible practices are followed in the mining and extraction of Li will become more important in the future. They describe the complexities in tracing Li through the supply chain, highlighting the need for finding a method that can trace Li back to its source. Lithium isotopes is a logical tool for this purpose. The paper would be of interest to scientists and other individuals interested in geochemistry, mining, and social and environmental accountability/responsibility.

Strengths of the paper include the compilation of existing data for the various sources of Li used in Li ion batteries from the literature. The manuscript also provides a detailed analysis of the effects on Li isotopic composition of the steps of the supply chain from source through metallurgy, cathode material synthesis and battery manufacturing.

A major weakness is the use of IQR values to describe the ranges of the different populations of data. This type of comparison minimizes the overlap among data populations and overrepresents differences. It is not an approach I have ever used, and I have not read any papers where this approach is used. If I understand it correctly, the IQRs take the data between the first quartile value and the third quartile value which means that fully half of the data is not considered when using the IQR range. The study would benefit from full use of the comprehensive set of data compiled. A more detailed analysis of the probability that a given sample or sample set is from a given source type or locality would give a more accurate idea of how useful Li isotopes will be in this endeavor. An analysis of how many samples would be required to distinguish sources would provide a clear picture of the utility of the method.

Additionally, the figures could be made a little more professional-looking. It is hard to put my finger on exactly what it is, but they are a bit jarring to the eye. Maybe it is the color choice of aqua and pinks that creates that feeling. The use of the company logos also makes it look more commercial than scientific. In Figure 3, it is hard to discern the data because the company logos are in the middle of the graph.

Below I discuss specific comments keyed to line and figure numbers in the manuscript draft.

Specific comments

Lines 74-91 - It would be more impactful to boldface the three sources using descriptive terms for each source. Rather than "The second source" and "The third source" boldface "Hard-rock lithium resources..." and "Sediment-hosted deposits..."

Lines 194-196 - The range of values considered for the salars appears to omit the data from the Qaidam Basin in China. Was this intentional? If so, this needs to be explained; if not, the ranges need to be recalculated including those samples.

Lines 215-218 - The manuscript states that the heavy isotope depletion of spodumene is confirmed by the data shown in Figure 2, saying that the spodumene values (-0.3 to 6.0‰) are "low and close" compared to UCC values (0+/-4‰). The comparison of IQR values for the spodumene data (50% of the data) with the 2sigma range for UCC (which would include 95% of the data) is not reasonable. Second, with an average value of +2.8‰, the spodumene are not low compared to UCC but are actually higher than the average value for UCC. Third, the range of values is 6.3‰ which is slightly more "close" than the range of 8‰ for UCC, but I am not sure it is worth highlighting. The data from figure 2 do clearly illustrate the final point of the paragraph, that there is greater variability in the d7Li values within a given location than there is between the different locations. I like that there is information included in the paragraph about relative isotopic fractionation among pegmatite minerals, however I think it is a stretch to say that the measured compositions illustrate this fractionation. I think the paragraph would be better written focusing on the full range of isotopic compositions for each location to describe this point more clearly.

Lines 223-229 - This case needs to be made with more nuance and with the full set of data, not just the IQR values. I am not convinced that the variations in composition measured are sufficient to demonstrate that one could discriminate among the different localities and among the different origins if one had an individual sample. It might be possible if you had a population of samples you could use statistical distributions to distinguish the origin. But as I see it, if you had a sample that falls in the region of overlap between salars and spodumene (with a d7Li between +2 and +12‰), it would be impossible to determine its origin. Similarly, for distinguishing locations, again if the sample happens to fall outside the overlap in composition it could be determined where it was from, but otherwise that would not be possible.

Lines 250-252 - Please define what is meant here by significant. Significant outside of analytical uncertainty?

Line 336 -The degree to which they are 'close' should be quantified in this sentence.

Lines 339-346 - It was unclear to me what the analyses of the sheets in the battery cell were meant to show. I understand that the compositions are homogeneous for the different sheets from the battery, but the point of that eludes me. Is there a significance to the composition of the sheets in this battery?

Lines 364-367 - This sentence is a bit unclear to me. How does the process-related fractionation of 5.5‰ allow the differentiation of ores produced using processes with different environmental or social impacts? Do the authors envision different amounts of fractionation produced by as yet unexplored processes? This should be explained.

Lines 370-373 - The study would benefit from a more quantitative approach. How high is the probability? The overlap region ("unknown origin domain") is understated because it is based on comparing just the IQR ranges. A full consideration of the data would show that the region of overlap encompasses a wider range of compositions.

Lines 530-534 - Sodium is an element that is commonly problematic in the analysis of Li isotopes, and therefore it is an important element to separate effectively. To that end, many authors report their methods in a way that documents the removal of Na from the sample. I don't see that discussed in the methods in this paper, and I think it would be beneficial for the authors to discuss this issue.

Figure 2 – The box-plot for the composition of the salars appears to not include the salars from China (Qaidam Basin), as the right-most value on the whiskers of the box, are less than 20‰, while the Qaidam Basin samples range up to 31‰. This needs to either be more clearly stated (why China is omitted) or else fixed to include the Chinese samples. The figure also shows probability ellipses, but nowhere is it stated what the probability of those ellipses is (75% of the samples? 90%?).

Figure 3 – This figure would benefit from clearer labeling and designation of different symbology used – both colors and shapes. There are small legends at the top of each subpart of the figure. The bottom two are fine, but the top one is confusing. Different colors are used to indicate form of Li (carbonate, hydroxide) but also to indicate different companies. There are some differences in shapes in this top part of the figure that do not have clear explanation. A more consistent use of color and shape would help convey the point more effectively. For example, it would be simpler to use color to indicate different forms of Li (carbonate, hydroxide, spodumene, analcime, etc.) and different symbol shapes to indicate different companies. Different symbols from those used in the top part could then be used in the bottom parts of the figure.

Reviewer #3 (Remarks to the Author):

Tracing the origin of lithium in Li-ion batteries using lithium isotopes

This is an interesting contribution, significant to the field and well documented therefore publication is recommended after some revision as follows:

In this manuscript the authors propose a method based on ^6Li and ^7Li isotope ratio for determining the origin of lithium in Lithium ion batteries in raw and processed materials, as 'fingerprints' for traceability and certification. They have analyzed relative isotopic ratios δ of different lithium sources, with low values (below +6 ‰) for hard rock and high values for brines (+11.9 ‰) and an unknown origin domain (between +6 ‰ and +11.9 ‰).

The authors also studied the relative isotopic ratios δ on prepared cathode materials and batteries, from different lithium sources and discuss the origin of lithium fractionation in the different steps of the production chain.

The paper is well written with full details of lithium mining and processing by different companies at different world locations.

However, there is not a discussion of the possibility of mixing salts from different origins and the high variability of lithium isotopic distribution in brines of South America Salars. Analysis of the lithium isotope distribution in the Lithium Triangle in South America shows that the value of δ varies between 4 and 12 for the salt flats of Atacama (Godfrey, 2020, Meixner 2021 and Munk 2018), Uyuni (Meixner 2021) and Olaroz (García 2020), Pozuelos (Meixner 2021), Hombre Muerto (Godfrey 2013), Centenario-Ratones (Orberger). These bibliography should be mentioned. The lithium fractioning depends on the type of rock source , and the temperature. In principle Uyuni and Atacama have higher values of δ (12-13) than Olaroz-Cauchari (5-8). With this in mind it is difficult for instance to trace the origin of lithium from Uyuni (Bolivia) and Atacama (Chile).

Please quote:

1. Lithium and Sr isotopic composition of salar deposits in the Central Andes across space and time: the Salar de Pozuelos, Argentina, Anette Meixner, Ricardo N. Alonso, Friedrich Lucassen, Laura Korte, Simone A. Kasemann, *Mineralium Deposita*, July 2021
doi: 10.1007/s00126-021-01062-3

2. Lithium and Lithium Isotopes in Earth's Surface Cycles

Philip A.E. Pogge von Strandmann, Simone A. Kasemann, and Josh B. Wimpenny
Elements, Vol. 16, pp. 253–258, (2020)
DOI: 10.2138/gselements.16.4.253

3. Lithium concentrations and isotope signatures of Palaeozoic basement rocks and Cenozoic volcanic rocks from the Central Andean arc and back-arc

Anette Meixner, Carisa Sarchi, Friedrich Lucassen, Raúl Becchio, Pablo J. Caffè, Jan Lindsay, Martin Rosner, Simone A. Kasemann, *Mineralium Deposita* (2020) 55:1071–1084.
<https://doi.org/10.1007/s00126-019-00915-2>.

Response to reviewers

My responses to the reviewers appear in blue in this document. The revisions appear also in blue in the manuscript.

Reviewer #1 (Remarks to the Author):

The paper is a very important contribution to the ongoing discussion on applicability of geochemical fingerprinting technologies to raw material supply chains. It is well written, well illustrated (except for Figure 4 which is too colourful and confusing to read- probably remove the flags on the right side), and clearly deserves to be published. The weakness is the limited data set for primary Li deposits, the strength is the approach, including an experimental approach, taken to investigate Li isotope fractionation. In the following, I will give some thoughts on the study and its potential applicability.

The study “Tracing the origin of lithium in Li-ion batteries using Li isotopes” describes a fingerprinting technology and concept for Li batteries. So far, it was not possible to receive information on the origin of lithium used in batteries. In the light of efforts to reach transparent and responsible supply chains for materials, this is a highly valuable first approach. It is especially important for those materials that are part of the “Green Deal”, where monitoring of CO₂ footprints for all process steps is essential; thus, the origin of all materials used in battery production must be monitored, from mining through processing into the battery factory and into the production of cars.

The authors use Li isotopes as the only tracing parameter. This has one major reason: processing Li ores and brines into intermediate materials used in battery production will likely destroy other tracing parameters such as trace element compositions. These are frequently used in fingerprinting studies for raw materials but usually do not “survive” the processing steps and are no longer indicative parameters once they leave the metallurgical processes. Therefore, it is a good approach to focus on the stable isotope composition of the major element in the mined ore (in this case, Li) and to investigate if this composition remains unaffected during any processing steps. This was done in the present study. The authors use two types of raw materials that, from a geological point of view, must reveal large differences in Li isotope composition. These differences have already been known from previous studies. Due to the limited number of Li isotope analyses from hard rock pegmatite deposits, which are one major supplier of Li via Li silicate (mostly spodumene), no major difference between such primary deposits could be found. However, when looking at the literature data on their Fig. S1, there is a wide spread that likely will discriminate different Li pegmatite deposits once more material for a better statistical evaluation has been analysed. The authors show on Fig. 2 that the interquartile range of $\delta^7\text{Li}$ in Li pegmatites varies from -0.3 to +6.0 permil, with individual analyses ranging from -3 to +12 permil (Fig. 1). $\delta^7\text{Li}$ of 1.1 to 4.4 permil in three spodumene concentrates analysed in the present study is in the expected range.

For the second important source of Li, brine deposits in South America and China, the authors

found a $\delta 7\text{Li}$ interquartile range of 7.2 to 11.9 permil in literature data, with individual data ranging from 2 up to 40 permil. The number of data points, especially for China, is not enough to calculate statistically valid numbers, resulting in high interquartile ranges. Nevertheless, evaluation of the literature data and the limited number of own data suggests that Li derived from pegmatites and salars can be discriminated using Li isotopes. But it must be kept in mind, that there is a significant overlap between pegmatite and salar Li that might render interpretation of $\delta 7\text{Li}$ values in the +2 to +12 permil range difficult because of possible overlap. Thus it is of utmost importance to develop a reliable set of data for each individual deposit from which Li is processed. Evaluation of the Li source is only possible if robust data on within-deposit variations are available; For pegmatite deposits, this should comprise repeated analyses of concentrates delivered to the plant for a certain period of time, e.g. within a year. It must also be kept in mind that Li isotope compositions may vary if further orebodies or other parts of – commonly zoned – pegmatites are being put in operation. In the case of Li from brines, repeated analyses of products must also be carried out; the large variation in Argentinean and Chinese deposits shown by literature data suggests inhomogeneity of Li isotopes within salars. The reason for this must be clarified.

In the present study, the authors put much emphasis on proving that Li isotope composition remains stable during purification. For pegmatites, calcination and sulphuric acid processing do not cause Li isotope fractionation. The new process developed by one company producing purified Li by ion exchange purification, however, leads to significant fractionation towards heavier isotope composition.

For Li produced from salars in Argentina, $\delta 7\text{Li}$ in Li carbonate is similar to the literature data from brines. Heavier isotope compositions measured in carbonate and hydroxide products from two companies using Salar de Atacama brines are explained by inadequate literature data, or by fractionation during extraction. This important point should be more carefully studied in the future.

Finally, cathode material synthesized from Li carbonate and hydroxide was analysed and it was shown that Li isotopes did not fractionate during synthesis.

The conclusions of the study are complex. On the one hand it can be shown that traditional processing of pegmatite-Li leaves light Li isotope compositions typical of the primary minerals and thus would make identification of Li origin in batteries possible. On the other hand the alternative extraction and purification process for pegmatite-Li leads to significantly heavier Li isotope composition, similar to brine-Li. The reason for unusually heavy compositions of Li salts produced from Atacama brines is not clear, but such heavy signatures are clearly not expected from pegmatite-Li. Therefore the authors tentatively split their data into three groups claiming that $\delta 7\text{Li} < 6$ permil most likely are derived from hard-rock pegmatite deposits, and values > 11.9 permil indicate a brine origin. All data in between fall into the “unknown origin domain”. These three categories provide a good starting point for the setup of a Li traceability process in the battery supply chain.

However, there are obvious weaknesses mainly due to inadequate sampling. If the data base will be improved in a way that robust data on Li isotope chemistry within the deposits currently producing Li are available, and final questions regarding fractionation during processing are resolved, the method has a high potential to be applicable in the future.

We have considered the specific comments made by the reviewer throughout the manuscript.

We revised the last sections of the paper dedicated to “*Assessing the geochemical traceability of lithium*” and “*Towards a methodological approach for certifying a responsible and sustainable lithium-supply chain*”, to highlight the fact that an evaluation of the Li source is only possible if robust data on within-deposit variations are available.

We added that the development of lithium certification is linked to the consumers’ interest in sustainable products, and Europe and the US are defending sustainable battery manufacturing projects.

We modified Figure 4 to make it less difficult to read e.g. by removing company logos and reducing flag size.

Reviewer #2 (Remarks to the Author):

General comments

This manuscript focuses on the utility of Li isotope measurement in tracing sources of lithium used in Li ion batteries. The authors point out that there is a growing demand for Li for use in such batteries and ensuring that both socially and environmentally responsible practices are followed in the mining and extraction of Li will become more important in the future. They describe the complexities in tracing Li through the supply chain, highlighting the need for finding a method that can trace Li back to its source. Lithium isotopes is a logical tool for this purpose. The paper would be of interest to scientists and other individuals interested in geochemistry, mining, and social and environmental accountability/responsibility.

Strengths of the paper include the compilation of existing data for the various sources of Li used in Li ion batteries from the literature. The manuscript also provides a detailed analysis of the effects on Li isotopic composition of the steps of the supply chain from source through metallurgy, cathode material synthesis and battery manufacturing.

A major weakness is the use of IQR values to describe the ranges of the different populations of data. This type of comparison minimizes the overlap among data populations and overrepresents differences. It is not an approach I have ever used, and I have not read any papers where this approach is used. If I understand it correctly, the IQRs take the data between the first quartile value and the third quartile value which means that fully half of the data is not considered when using the IQR range. The study would benefit from full use of the comprehensive set of data compiled. A more detailed analysis of the probability that a given sample or sample set is from a given source type or locality would give a more accurate idea of how useful Li isotopes will be in this endeavor. An analysis of how many samples would be required to distinguish sources would provide a clear picture of the utility of the method.

Additionally, the figures could be made a little more professional-looking. It is hard to put my

finger on exactly what it is, but they are a bit jarring to the eye. Maybe it is the color choice of aqua and pinks that creates that feeling. The use of the company logos also makes it look more commercial than scientific. In Figure 3, it is hard to discern the data because the company logos are in the middle of the graph.

The literature data available in previous papers are dedicated to understanding the formation of Li deposits; they are a collection of various samples containing highly variable levels of lithium (from 0.07 to 5000 mg/L for salars). Among these samples, it is difficult to know what is actually processed for transformation into lithium salt. This is why we used here the interquartile range (IQR) of Li isotope compositions as a measure of statistical dispersion to describe the ranges of the different populations of data. Following the reviewer's comments, we separated the salar data into two sets (China and "Li triangle") to be more statistically significant. The IQR values (and box-plot representation) highlight the statistical difference between hard-rock and brine due to the physico-chemical conditions of ore-forming processes, which was already predicted by *ab initio* calculations. This result provides a good starting point for the setup of a Li traceability process in the battery supply chain, and our paper lays the foundation of a method based on lithium isotopes for tracing the origin of lithium in batteries. However, as explained in the last part of the paper, the method parameters will need to be refined as new data are acquired on the different deposits exploited and the different extraction processes used. It will be necessary to collaborate with lithium-salt producers to evaluate the signature of their products, because these signatures will be found in the batteries. A reference database with comprehensive up-to-date data on available Li products and a specific statistical data evaluation strategy must also be developed.

We modified all the figures of the manuscript to make them more professional-looking, such as by removing company logos.

Below I discuss specific comments keyed to line and figure numbers in the manuscript draft.

Specific comments

Lines 74-91 - It would be more impactful to boldface the three sources using descriptive terms for each source. Rather than "The second source" and "The third source" boldface "Hard-rock lithium resources..." and "Sediment-hosted deposits..."

Lines 81-87 - This change was made.

Lines 194-196 – The range of values considered for the salars appears to omit the data from the Qaidam Basin in China. Was this intentional? If so, this needs to be explained; if not, the ranges need to be recalculated including those samples.

Lines 181-187 - As shown on Fig. S1 (previous version), the value range between +7.2 to +11.9‰ for salars includes the data from the Qaidam Basin in China. However, as very few data are available in the literature for China (n=20) compared to South American salars (n=103 in the new draft), this graphic representation was not suitable. Following the reviewer's comments, we

show on the figures of the new MS version a box plot for China (+16.1 to +31.4‰) and another for “Li triangle” salars (+7.9 to +11.3‰) to better appreciate the variability of salar signatures. The text has been amended to take account of this change.

Lines 215-218 – The manuscript states that the heavy isotope depletion of spodumene is confirmed by the data shown in Figure 2, saying that the spodumene values (-0.3 to 6.0‰) are “low and close” compared to UCC values (0+/-4‰). The comparison of IQR values for the spodumene data (50% of the data) with the 2sigma range for UCC (which would include 95% of the data) is not reasonable. Second, with an average value of +2.8‰, the spodumene are not low compared to UCC but are actually higher than the average value for UCC. Third, the range of values is 6.3‰ which is slightly more “close” than the range of 8‰ for UCC, but I am not sure it is worth highlighting. The data from figure 2 do clearly illustrate the final point of the paragraph, that there is greater variability in the $\delta^{7}\text{Li}$ values within a given location than there is between the different locations. I like that there is information included in the paragraph about relative isotopic fractionation among pegmatite minerals, however I think it is a stretch to say that the measured compositions illustrate this fractionation. I think the paragraph would be better written focusing on the full range of isotopic compositions for each location to describe this point more clearly.

Lines 226-227 -We modified the sentence to clarify this point in the text.

Lines 223-229 – This case needs to be made with more nuance and with the full set of data, not just the IQR values. I am not convinced that the variations in composition measured are sufficient to demonstrate that one could discriminate among the different localities and among the different origins if one had an individual sample. It might be possible if you had a population of samples you could use statistical distributions to distinguish the origin. But as I see it, if you had a sample that falls in the region of overlap between salars and spodumene (with a $\delta^{7}\text{Li}$ between +2 and +12‰), it would be impossible to determine its origin. Similarly, for distinguishing locations, again if the sample happens to fall outside the overlap in composition it could be determined where it was from, but otherwise that would not be possible.

In our section “*Isotope variability between lithium deposits and among coexisting ores*”, we introduced some nuances to our conclusions.

The goal of our paper is to demonstrate the potential of Li isotope tools for assessing the geochemical traceability of lithium. As debated in sections “*Assessing the geochemical traceability of lithium*” and “*Towards a methodological approach for certifying a responsible and sustainable lithium-supply chain*”, this will need the further development of a reference database as well as a specific statistical data evaluation strategy.

Lines 250-252 – Please define what is meant here by significant. Significant outside of analytical uncertainty?

Lines 261-262- We clarified this point in the text.

Line 336 –The degree to which they are ‘close’ should be quantified in this sentence.

Line 351- We clarified this point in the text.

Lines 339-346 – It was unclear to me what the analyses of the sheets in the battery cell were meant to show. I understand that the compositions are homogeneous for the different sheets from the battery, but the point of that eludes me. Is there a significance to the composition of the sheets in this battery?

Lines 361-367- We analysed various sheets of a same battery to verify that we have a single $\delta^7\text{Li}$ value within a battery. This homogeneity of composition is a pre-requisite if we want to use Li isotope compositions to trace the origin of lithium in batteries. We clarified this point in the text.

Lines 364-367 – This sentence is a bit unclear to me. How does the process-related fractionation of 5.5‰ allow the differentiation of ores produced using processes with different environmental or social impacts? Do the authors envision different amounts of fractionation produced by as yet unexplored processes? This should be explained.

Lines 389-391- We have clarified this point in the text by adding an example.

Lines 370-373 – The study would benefit from a more quantitative approach. How high is the probability? The overlap region (“unknown origin domain”) is understated because it is based on comparing just the IQR ranges. A full consideration of the data would show that the region of overlap encompasses a wider range of compositions.

Lines 392-396- We added in the text that this is a first estimation to demonstrate the potential of the method. As mentioned in the last part of the article, in the future an evolving database and a specific statistical data evaluation strategy will be necessary for assessing the geochemical traceability of lithium.

Lines 530-534 – Sodium is an element that is commonly problematic in the analysis of Li isotopes, and therefore it is an important element to separate effectively. To that end, many authors report their methods in a way that documents the removal of Na from the sample. I don't see that discussed in the methods in this paper, and I think it would be beneficial for the authors to discuss this issue.

Sodium is a problematic element when analysing Li isotopes. The chemical procedure for Li purification used in this study was described and validated by a previous paper of the BRGM team (Millot et al., 2004). They verified the chemical separation of Li from matrix by using the reference value of a seawater sample (IRMM BCR-403). In the present paper, the removal of Na is not specifically discussed as only the analcine sample contains a notable quantity of Na; the

other samples: spodumene concentrate, purified chemical products (hydroxide and carbonate) and cathode active material contain only traces of sodium.

Figure 2 – The box-plot for the composition of the salars appears to not include the salars from China (Qaidam Basin), as the right-most value on the whiskers of the box, are less than 20‰, while the Qaidam Basin samples range up to 31‰. This needs to either be more clearly stated (why China is omitted) or else fixed to include the Chinese samples. The figure also shows probability ellipses, but nowhere is it stated what the probability of those ellipses is (75% of the samples? 90%?).

We have modified this figure to better appreciate the variability of salars signatures, we represented a box plot for China (+16.1 to +31.4‰) and another for “Li triangle” salars (7.9 to +11.3‰).

The confidence level for ellipses is also added in the Figure 2 caption (confidence level $p=0.68$).

Figure 3 – This figure would benefit from clearer labeling and designation of different symbology used – both colors and shapes. There are small legends at the top of each subpart of the figure. The bottom two are fine, but the top one is confusing. Different colors are used to indicate form of Li (carbonate, hydroxide) but also to indicate different companies. There are some differences in shapes in this top part of the figure that do not have clear explanation. A more consistent use of color and shape would help convey the point more effectively. For example, it would be simpler to use color to indicate different forms of Li (carbonate, hydroxide, spodumene, analcime, etc.) and different symbol shapes to indicate different companies. Different symbols from those used in the top part could then be used in the bottom parts of the figure.

We modified Figure 3 to make it less difficult to read, by removing the company logos and changing the small legends at the top of each subpart of the figure.

Reviewer #3 (Remarks to the Author):

Tracing the origin of lithium in Li-ion batteries using lithium isotopes

This is an interesting contribution, significant to the field and well documented therefore publication is recommended after some revision as follows:

In this manuscript the authors propose a method based on ^6Li and ^7Li isotope ratio for determining the origin of lithium in Lithium ion batteries in raw and processed materials, as ‘fingerprints’ for traceability and certification. They have analyzed relative isotopic ratios δ of different lithium sources, with low values (below +6 ‰) for hard rock and high values for brines (+11.9 ‰) and an unknown origin domain (between +6 ‰ and +11.9 ‰).

The authors also studied the relative isotopic ratios δ on prepared cathode materials and batteries, from different lithium sources and discuss the origin of lithium fractionation in the different steps of the production chain.

The paper is well written with full details of lithium mining and processing by different companies at different world locations.

However, there is not a discussion of the possibility of mixing salts from different origins and the high variability of lithium isotopic distribution in brines of South America Salars. Analysis of the lithium isotope distribution in the Lithium Triangle in South America shows that the value of δ varies between 4 and 12 for the salt flats of Atacama (Godfrey, 2020, Meixner 2021 and Munk 2018), Uyuni (Meixner 2021) and Olaroz (García 2020), Pozuelos (Meixner 2021), Hombre Muerto (Godfrey 2013), Centenario-Ratones (Orberger). These bibliography should be mentioned.

The lithium fractioning depends on the type of rock source , and the temperature. In principle Uyuni and Atacama have higher values of δ (12-13) than Olaroz-Cauchari (5-8). With this in mind it is difficult for instance to trace the origin of lithium from Uyuni (Bolivia) and Atacama (Chile).

Please quote:

1. Lithium and Sr isotopic composition of salar deposits in the Central Andes across space and time: the Salar de Pozuelos, Argentina, Anette Meixner, Ricardo N. Alonso, Friedrich Lucassen, Laura Korte, Simone A. Kasemann, Mineralium Deposita, July 2021
doi: 10.1007/s00126-021-01062-3

2. Lithium and Lithium Isotopes in Earth's Surface Cycles

Philip A.E. Pogge von Strandmann, Simone A. Kasemann, and Josh B. Wimpenny
Elements, Vol. 16, pp. 253–258, (2020)
DOI: 10.2138/gselements.16.4.253

3. Lithium concentrations and isotope signatures of Palaeozoic basement rocks and Cenozoic volcanic rocks from the Central Andean arc and back-arc

Anette Meixner, Carisa Sarchi, Friedrich Lucassen, Raúl Becchio, Pablo J. Caffè, Jan Lindsay, Martin Rosner, Simone A. Kasemann, Mineralium Deposita (2020) 55:1071–1084.

<https://doi.org/10.1007/s00126-019-00915-2>.

We modified the paragraph "*Isotope variability between lithium deposits and among coexisting ores*" to highlight the strong variability of lithium isotopic distribution in brines of South American salars. Moreover, some clarifications are given on the origin of the lithium fractioning in salars.

An active materials manufacturer can be supplied with lithium salts from different origins (see paragraph "*Assessing the geochemical traceability of lithium*"), but these different salts will not be mixed within the same active material. However, producers of lithium salts can exploit different deposits with various signatures, so it will be necessary to amend the traceability database as new deposits come on stream. We added this point in the paragraph "*Towards a methodological approach for certifying a responsible and sustainable lithium-supply chain*".

We added missing references to previous articles on South American salars in this manuscript (Meixner et al., 2022, 2020; Munk et al., 2018; Pogge von Strandmann et al., 2020). In particular, we used data available in Meixner et al. (2022) and Munk et al. (2018) to improve the data on the Bolivia, Chile and Argentina salars (in the text, and on figures 2, 3, 4 and S1).

References

- Meixner, A., Alonso, R.N., Lucassen, F., Korte, L., Kasemann, S.A., 2022. Lithium and Sr isotopic composition of salar deposits in the Central Andes across space and time: the Salar de Pozuelos, Argentina. *Miner. Depos.* 57, 255–278. <https://doi.org/10.1007/s00126-021-01062-3>
- Meixner, A., Sarchi, C., Lucassen, F., Becchio, R., Caffè, P.J., Lindsay, J., Rosner, M., Kasemann, S.A., 2020. Lithium concentrations and isotope signatures of Palaeozoic basement rocks and Cenozoic volcanic rocks from the Central Andean arc and back-arc. *Miner. Depos.* 55, 1071–1084. <https://doi.org/10.1007/s00126-019-00915-2>
- Millot, R., Guerrot, C., Vigier, N., 2004. Accurate and high-precision measurement of lithium isotopes in two reference materials by MC-ICP-MS. *Geostand. Geoanalytical Res.* 28, 153–159. <https://doi.org/10.1111/j.1751-908X.2004.tb01052.x>
- Munk, L.A., Boutt, D.F., Hynek, S.A., Moran, B.J., 2018. Hydrogeochemical fluxes and processes contributing to the formation of lithium-enriched brines in a hyper-arid continental basin. *Chem. Geol.* 493, 37–57. <https://doi.org/10.1016/j.chemgeo.2018.05.013>
- Pogge von Strandmann, P.A.E., Kasemann, S.A., Wimpenny, J.B., 2020. Lithium and lithium isotopes in earth's surface cycles. *Elements* 16, 253–258. <https://doi.org/10.2138/GSELEMENTS.16.4.253>

REVIEWERS' COMMENTS

Reviewer #2 (Remarks to the Author):

Overall, in its revised format, the manuscript does a good job of illustrating the potential and limitations of using Li isotopes as a fingerprint. The changes made by the authors more clearly explain their ideas and also better articulate limitations of the dataset. I think breaking out the China dataset from the other salars as they have done more clearly shows the possibilities that exist for differentiating Li sources from some localities. While it still seems to me that IQR is not an ideal representation of the data, with the revisions in associated text it is more palatable in the current version of the manuscript. I don't understand the comment in their rebuttal about not knowing the sources of the materials in the samples compiled from the literature. The point of their study is that using this method you should be able to discern source localities regardless of original source – that the isotopic composition is mostly retained. In this context, it seems strange for them to say that they don't want to use statistical methods because the samples of unknown origin.

I still think it is a huge challenge for implementation of this method that there is large range of isotopic compositions which cannot be "fingerprinted." I think that this aspect needs to be acknowledged in the final section "Towards a methodological approach..." The literature compilation suggests that samples from at least half of the localities (Australian pegmatites, most of the South American salars) fall in the isotopic range for which there is uncertainty of source.

Reviewer #3 (Remarks to the Author):

I am satisfied with the revised manuscript.

REVIEWERS' COMMENTS

My responses to the reviewers appear in blue in this document. The revisions appear also in blue in the manuscript.

Reviewer #2 (Remarks to the Author):

Overall, in its revised format, the manuscript does a good job of illustrating the potential and limitations of using Li isotopes as a fingerprint. The changes made by the authors more clearly explain their ideas and also better articulate limitations of the dataset. I think breaking out the China dataset from the other salars as they have done more clearly shows the possibilities that exist for differentiating Li sources from some localities. While it still seems to me that IQR is not an ideal representation of the data, with the revisions in associated text it is more palatable in the current version of the manuscript. I don't understand the comment in their rebuttal about not knowing the sources of the materials in the samples compiled from the literature. The point of their study is that using this method you should be able to discern source localities regardless of original source – that the isotopic composition is mostly retained. In this context, it seems strange for them to say that they don't want to use statistical methods because the samples of unknown origin.

I still think it is a huge challenge for implementation of this method that there is large range of isotopic compositions which cannot be "fingerprinted." I think that this aspect needs to be acknowledged in the final section "Towards a methodological approach..." The literature compilation suggests that samples from at least half of the localities (Australian pegmatites, most of the South American salars) fall in the isotopic range for which there is uncertainty of source.

We added the limitation of the method in the final section "Towards a methodological approach for certifying a responsible and sustainable lithium-supply chain".

••••

Reviewer #3 (Remarks to the Author):

I am satisfied with the revised manuscript.

Great!